# Occupational Safety, Health, and Well-Being Concerns and Solutions for Management Reported by Sign Language Interpreters: A Qualitative Study

**DOI:** 10.3390/ijerph21111400

**Published:** 2024-10-23

**Authors:** Gretchen Roman, Cristina Demian, Tanzy Love, Reza Yousefi-Nooraie

**Affiliations:** 1Department of Family Medicine, University of Rochester, 1381 South Avenue, Rochester, NY 14620, USA; 2Department of Environmental Medicine, University of Rochester, 400 White Spruce Boulevard, Rochester, NY 14623, USA; cristina_demian@urmc.rochester.edu; 3Department of Biostatistics and Computational Biology, University of Rochester, 265 Crittenden Boulevard, Rochester, NY 14642, USA; tanzy_love@urmc.rochester.edu; 4Department of Public Health Sciences, University of Rochester, 265 Crittenden Boulevard, Rochester, NY 14642, USA; reza_yousefi-nooraie@urmc.rochester.edu

**Keywords:** mental health, organizational culture, physical health, sign language interpreters, *Total Worker Health^®^*

## Abstract

While the occupational health of sign language interpreters has traditionally focused on physical health, evidence demonstrating mental health concerns is growing and supporting a shift to a more integrated approach. We embarked on a qualitative study to guide the adaptation of a previously developed *Total Worker Health^®^* program to the context of sign language interpreting. Eight unstructured 90-min focus groups were conducted. Interpreters reported occupational safety, health, and well-being concerns and shared their solutions for management. Twenty-seven interpreters participated (aged 53.7 years; 81% female; 85% white). Predominant concerns centered on topics like workplace violence, secondary traumatic stress or vicarious trauma, lack of work–life integration or boundaries, and loss of agency or loss of self. The organizational culture of the field fostered deprioritization of self, oppression, elitism, sexism, and unhealthy relationships with interpreter peers and community members. Physical health remained a contributor, specifically the physical effects of non-physical work, aging, and differences in exposures across interpreting settings but paled in comparison to mental health and organizational culture. Solutions for management included but were not limited to prioritization of jobs, creating safe spaces/communities of supported practice, and exercise. This study supports the evolution of future research, practice, policy, and capacity building aspiring to a more integrated approach.

## 1. Introduction

Because it is estimated that there are 522,000 adults in the United States who range from moderate trouble hearing to Deaf and use sign language to communicate [1] and 10,385 certified members in the Registry of Interpreters for the Deaf (RID) [2], the national professional association for sign language interpreters, the ratio of Deaf sign language users to interpreters is fifty to one. Communication access is a social determinant of health in Deaf communities [3]; subsequently, there is a heightened concern regarding the many variables that impact the occupational health of sign language interpreters.

Traditionally, much of the previous literature investigating the occupational health of sign language interpreters has focused on physical health, such as overuse syndromes [4,5,6], cumulative trauma [7], and musculoskeletal disorders [8,9,10]. Evidence has indicated that an interpreter’s physical exertion [11], ergonomics, posture, and workstation set-up [12], and sign production and style or upper limb biomechanics [7,13,14,15,16] may contribute to such physical health concerns. Sign language interpreting has been identified as a practice profession [17]. When compared with other practice professions, such as teaching, medicine, and law enforcement, interpreters reported significantly more physical exertion [11]. Roman and Samar (2015) [12] found that improved ergonomics knowledge, postural awareness, and workstation setup reduced interpreters’ reports of musculoskeletal pain. “Micro” rest breaks, muscle tension, hand and wrist deviations from neutral, ballistic signing (unnecessary abruptness and force), and work envelope or sign space were identified as unique to sign language interpreters’ biomechanics [7,13]. Qin et al. (2008) [18] found that movements while signing in interpreters’ dominant wrists were 1.3 to 1.5 times greater than the established high-risk industrial benchmarks. While interpreters’ biomechanics were able to explain 30% of the variance in reported pain, Roman et al. (2021b) [19] were left wondering, what other elements contribute to the remaining 70%? Perhaps, these other elements relate to the mental health concerns of interpreters.

Evidence about the mental health concerns of sign language interpreters has been growing; particularly, the report of stress [7,9,18,20,21,22,23], burnout [21,23,24,25,26], mental and psychosocial fatigue [27], and secondary traumatic stress or vicarious trauma [23,25,26,28,29,30]. At baseline, the prevalence of interpreters reporting moderate or greater perceived stress, burnout, and secondary traumatic stress ranged from 41% to 76% [23]. Qin et al. (2008) [18] found that stress increased the velocity and acceleration of the non-dominant wrist in interpreters. As seen with physical exertion, when compared with other practice professions, interpreters also reported significantly more psychological distress and depression [11]. Working interpreters with high stress have demonstrated dysregulated cortisol levels and such changes have been associated with negative long-term health consequences [22]. Other work pointed out that interpreters are at high risk of exposure to traumatic events and that the field is lacking in the necessary preventive strategies and support systems [28]. An additional handful of studies found emerging mental health concerns, such as horizontal violence and emotional or psychological safety [31,32].

There is support for incorporating the many variables that impact the physical and mental health of interpreters into a more integrated approach. In Woodcock and Fisher’s (2008) [33] proposed conceptual model of interpreter injury, they not only identified sign production and style, which included movement frequency, joint angle, and force but also elements of the interpreting situation and interpreters’ individual characteristics as relevant attributes to interpreters’ injury-risk development. The interpreting situation included the psychosocial elements of the event and the interpreter’s state of mind, whereas the interpreter’s individual characteristics included individual physiology and behavior, physical tension, and fitness. Fisher et al. (2012a) [9] also created a conceptual model of musculoskeletal disorder development based on findings from a systematic review. They identified mechanical exposure, which includes increased postural deviations and sign velocity or acceleration, along with the speaker’s pace as a contributor to movement rate; however, they also recognized the role of psychosocial and environmental stress as a non-mechanical contributor to musculoskeletal pathology [9]. In addition, more recent work conducted during the COVID-19 pandemic supported this evolution to a more integrated approach. Upon investigating the occupational safety and health of sign language interpreters working remotely, Roman et al. (2022a) [34] found a strong positive association between the outcomes measuring physical and mental health.

Established by the National Institute for Occupational Safety and Health at the Centers for Disease Control and Prevention in 2011, *Total Worker Health^®^* (TWH) offers an excellent integrated framework for addressing the occupational health concerns of sign language interpreters. TWH is demonstrated by research, practice, policy, and capacity building that combines protection from work-related hazards with injury and illness prevention to advance worker well-being. It expands the traditional delivery of occupational safety and health to the delivery of occupational safety, health, and well-being [35,36,37]. TWH has five defining elements: (1) demonstrate leadership commitment to worker safety and health at all levels of the organization, (2) design work to eliminate or reduce safety and health hazards and promote worker well-being, (3) promote and support worker engagement throughout program design and implementation, (4) ensure confidentiality and privacy of workers, and (5) integrate relevant systems to advance worker well-being [38]. Developed to decrease the fragmentation of occupational health services offered to workers, TWH is a comprehensive approach that aims to address primary and secondary prevention, physical and mental health, individual- and organizational-level factors, as well as the intersection of work and non-work. Having safe and healthful working conditions is a fundamental principle of TWH [37]. In other words, when workers feel physically and emotionally or psychologically safe at work, they are healthier.

In anticipation of adapting a previously developed TWH program [39,40], we embarked on a qualitative study to supplement the findings from our review of the literature. This work fell within the explore and plan phases of the TWH program lifecycle. We raised awareness about integrative efforts to improve worker well-being, shared previously identified attributes relating to interpreters’ injury-risk development, and then assessed interpreters’ needs by gathering insights from their work-related experiences. Our aim was to create an open forum for sign language interpreters to express their occupational safety, health, and well-being concerns, as well as share their solutions for management.

## 2. Materials and Methods

This qualitative study (STUDY00008720) underwent review by the University of Rochester’s Research Subjects Review Board. Based on federal and University criteria, it was deemed exempt and, thus, the requirement of written informed consent from participants was waived. We adhered to the Standards for Reporting Qualitative Research (SRQR; Appendix A) [41].

### 2.1. Participants

Adults 18 years or older were eligible to participate if they maintained certification as sign language interpreters bilingual in English and American Sign Language. We aspired for purposive maximum variation (demographic and geographic) sampling. Leadership of all the affiliate chapters across each of the five regions (Northeast, Southeast, Midwest, Central, and Pacific) of RID and specific member section (e.g., Interpreters and Transliterators of Color Member Section) leadership of RID were contacted and asked to advertise participation in the study by sharing/posting the provided recruitment flyer via social media, websites, email listservs, and/or displaying flyers at various locations to actively recruit sign language interpreters within their regions. A minimum number of interpreting hours per week or years of interpreting experience were not required. The recruitment flyer and information about participation in this study were also emailed directly to interpreters who identify as male and/or Deaf who publicly shared their contact information on RID’s website.

In Roman et al. (2023b), no new topic areas, themes, or sub-themes emerged after completing five focus groups with interpreters (n = 22) and five individual interviews with interpreting administrators (n = 5) about the positive and negative consequences of transitioning from onsite to remote interpreting in response to the COVID-19 pandemic. Thus, similar to the previous work [42], we aimed to recruit between 25 and 30 participants in this study.

### 2.2. Data Collection

Eight unstructured 90-min focus groups were conducted between December 2023 and March 2024. One focus group occurred across each of the five regions of RID, as well as one with English and American Sign Language interpreters based outside of the United States and two with certified Deaf interpreters. Because the one investigator who facilitated all of the qualitative data collection (GR) also maintains her certification in interpretation, certification in transliteration, and national interpreter certification as a certified sign language interpreter, six focus groups with interpreters who all identified as hearing or non-Deaf were completed in spoken English and two focus groups with interpreters who all identified as Deaf were completed in sign language. Groups were performed virtually and recorded using video conferencing software (Zoom, v. 5.17, San Jose, CA, USA).

The first fifteen minutes of each session welcomed participants, allowed for introductions, explained study procedures and expectations, and provided the opportunity to ask any questions. Then, for the second fifteen minutes, the facilitator briefly shared background information relevant to TWH [35,36,37] and the model of interpreter injury [33]. The remaining time allowed for the interpreters to organically share their occupational safety, health, and well-being concerns and dialogue about their solutions for management. No question guide was used.

### 2.3. Data Processing and Analysis

One investigator (GR) listened to the audio recordings from the unstructured focus groups completed in spoken English, and then manually transcribed them into English. The same investigator (GR) viewed the video recordings from the groups completed in sign language, then simultaneously interpreted and manually transcribed them into English. Rapid qualitative analysis of each focus group occurred sequentially and information gleaned was used to inform subsequent groups [43]. Transcript summaries were created by the same investigator who collected and transcribed the data (GR) to reduce the data into domains and then a summary matrix was established to compile the data across focus groups. Another investigator (RY-N) audited the system of documentation that was used to reduce, compile, reconstruct, and synthesize the raw data [44,45]. Two investigators (GR,RY-N) then worked together, reaching a consensus on the domains relevant to the identified occupational safety, health, and well-being concerns and management solutions reported by interpreters. Saturation was determined to occur when groups began reiterating similar information gleaned from previous groups without any prompting.

## 3. Results

### 3.1. Participants

Twenty-seven interpreters residing in four different countries participated (Table 1). Nineteen interpreters participated in the focus groups across the Northeast (n = 4), Southeast (n = 4), Midwest (n = 4), Central (n = 3), and Pacific (n = 4) regions of RID. Four interpreters participated in the focus group with interpreters based outside of the United States and four interpreters (or two interpreters in each) participated in the two certified Deaf interpreter focus groups. Within the United States, 20 different states were represented. For the interpreters based outside of the United States, we can share they were from the continents of North America, South America, and Europe but are not disclosing their specific countries to preserve their anonymity. Interpreters represented a wide distribution of interpreting settings (e.g., community freelance, which indicates that the interpreter works as an independent contractor or is self-employed), interpreting certifications, and levels of education. Eighty-five percent of the interpreters were white and 82% identified as hearing and female. Although skewed toward interpreters who are white and female, the demographics of this sample were consistent with the previous reports of the sign language interpreting field being between 85–87% white and 75–82% female [34,46,47,48]. The 19% representation of Deaf and hard-of-hearing interpreters in this study was greater than the 1% previously reported [34,48].

### 3.2. Sign Language Interpreters’ Familiarity with Occupational Health

Sign language interpreters expressed varying degrees of familiarity with occupational health. Some had no familiarity, “This is the first time I have ever heard anyone talk about occupational health in relation to sign language interpreting. We don’t talk about it, at least in my community (Interpreter #1)”. Other interpreters were familiar but not with its application to the field of sign language interpreting, “I am familiar with OSHA [Occupational Safety and Health Administration], but things like black lung disease and mining, people who are in physically dangerous [signed quotes] occupations and I have thought of OSHA in those scenarios… factories, even repetitive motion, but more like assembly lines. I don’t think I’ve heard it very much in relationship to interpreting (Interpreter #15)”. Another interpreter said, “I’ve heard of that [occupational health], but I suppose if you asked me to define it, I wouldn’t necessarily be able to other than just making a best guess about it. I certainly have not really heard it in terms of the field of interpreting. What I have more heard is talking about burnout prevention or self-care in this broad and nebulous way. I’ve seen professional development workshops about burnout and self-care, but it feels very much like it’s not realistically framed for fitting into an interpreter’s actual life (Interpreter #21)”.

Some knew about previous work relating to occupational health in the field of sign language interpreting but were not familiar with any more recent efforts, “I have heard about a lot of the work on occupational health and safety. I attended a RID Conference, and they were doing work on the physical part of what you can do… that was 15, almost 20 years ago (Interpreter #2)”. Whereas others were aware that people were talking about occupational health as it pertains to the field but that there needed to be more of an emphasis and that “…we, as a community, could be more well (Interpreter #9)”. “I started out interpreting in a vocational school and nobody was talking about occupational health. They would send you to interpret a two-and-a-half-hour class alone. You’re young, you’re stupid, you’re healthy, and nothing hurts, so why not? It honestly wasn’t until I started working at VRS [video relay service], which was 20 years into my career that I first heard someone talking about occupational health and that was really more about making sure you take your break every 50 min but nobody was explaining the why behind it or emphasizing that it’s for your mental health, it’s for your physical health… so, I don’t think, in this field, there’s a lot of emphasis on it at all (Interpreter #19)”.

### 3.3. Occupational Safety, Health, and Well-Being Concerns of Sign Language Interpreters

#### 3.3.1. Mental Health

Mental health, by far, was the most prominent domain expressed within the occupational safety, health, and well-being concerns of sign language interpreters (Figure 1; Appendix A). “I feel like the thing that I have seen most is not so much connected to the physical strain of the work but the emotional and psychological demands (Interpreter #15)”. Workplace violence, secondary traumatic stress or vicarious trauma, lack of work–life integration or boundaries, loss of agency or loss of self, and isolation were identified as sub-domains of mental health. Workplace violence was pervasive, including horizontal violence or bullying (negative relationships between hearing sign language interpreters and negative relationships between Deaf sign language interpreters), vertical violence (negative relationships between hearing and Deaf sign language interpreters, as well as between sign language interpreters and Deaf community members), and interpreting agency violence toward sign language interpreters. When Deaf and hearing interpreters team together for an assignment, one Deaf interpreter shared, “Yes, if I team with a hearing interpreter, in theory, from a professional perspective, the two of us are horizontal but when you think about the system of power, it is not equitable, so it becomes vertical violence (Interpreter #25)”. Workplace violence was compounded if the interpreter identified as a member of the neurodiverse and/or lesbian, gay, bisexual, transgender, and queer or questioning (LGBTQ) communities. Regarding secondary traumatic stress or vicarious trauma, an interpreter commented, “I think interpreters have such a unique position and a unique job in that we are every part of people’s lives. So, I know for myself, I struggle a lot with the mental health part of interpreting. Throughout the past 10 years, I have learned that I don’t do as well with the really intense emotional assignments because I take that on and then, that affects me the rest of the day or depending on how horrible the situation was, maybe for the next week (Interpreter #1)”.

Regarding the lack of work–life integration or boundaries and loss of agency or loss of self sub-domains, interpreters described pieces of themselves that were missing and losing a sense of themselves outside of their professional identity: “I was fortunate to participate in a workshop that forced me to really examine, who am I outside of interpreting? I’m not an interpreter. Interpreting is what I do to earn a living, but I’m so much more than an interpreter and so, I had to really take a look at who am I? If I am not interpreting, what else would I be? What else could I do? And so, I have other outlets, as well. I’m a pet Mom. I love to play in the kitchen. I love to write, so I have all of these other options that if I had to give up interpreting, I wouldn’t feel like I’m dead now. I have nothing left. I have all of these other interests that I can pursue. Who are you outside of being an interpreter, I think, is really, really important (Interpreter #19)”.

There were some differences in psychological exposures noted across interpreting settings; specifically, in video remote interpreting, VRS interpreting, and community interpreting or freelance. In general, interpreters with extralinguistic knowledge or knowledge about the topic being discussed during the interpreting assignment were felt to experience better occupational health or less stress than those with limited subject matter knowledge.

#### 3.3.2. Organizational Culture

The organizational culture of the field was the second largest expressed contributor to the occupational health of interpreters (Figure 1; Appendix A). One interpreter explained, “It’s a badge of honor that you haven’t slept, you’ve only drank coffee all day long, and you’re in pain but you’re going to keep doing this job because you have to provide access. At the expense of what? That’s my question (Interpreter #18)”. And another interpreter responded, “Even when there is more of a focus on people leaving the field, do we have a culture as sign language interpreters, is there something that we’re picking up somewhere that we will just continue to do anyway? Like [Interpreter #18] was talking about, almost having that badge of honor, this is how hard I’ve worked, I’ve done this much myself, and I’ve had this many coffees today, even if we had more of the structural changes, is there more of a culture that needs to shift, as well (Interpreter #20)?”

Deprioritization of self, system or organizational challenges, oppression, “isms”, relationships with interpreter peers and community members, and not being seen as equivalent professionals were identified as sub-domains of organizational culture. The deprioritization of self sub-domain overlaps some with the loss of agency or loss of self noted in the mental health domain. System or organizational challenges included “uberization” of the field, “…we’re promoting this idea, and not to blame VRS, but I think we’re entraining the community, users, people who purchase our services that you can have an interpreter at any point in time that you choose to have one. We’re entraining that into the psyche of our customers and people who purchase the service, who are on the hook to provide ADA [Americans with Disabilities Act], to follow the law. And Deaf people, who now think, “oh, I don’t need to wait and I’m so used to [signed snap, snap, snap], why can’t I have an interpreter right now?” That is going to be to our peril. And, as a consequence to what I call the uberization, so as a company having to respond to these external demands, then interpreters just need to be ready at a moment’s notice, to be at a beck and call and there is no return loyalty or commitment to interpreters by the people who are hiring them (Interpreter #6)”.

The oppression, “isms,” and relationships with interpreter peers and Deaf community members sub-domains all overlap some with the secondary traumatic stress or vicarious trauma and workplace violence noted in the mental health domain. The oppression sub-domain highlighted the systemic oppression that sign language interpreters witness when working with Deaf community members, as well as the oppression and subjugated professional status experienced by black, indigenous, and other people of color (BIPOC), LGBTQ, and Deaf interpreters. “Isms” were noted among Deaf interpreters; specifically elitism and sexism. Relationships with interpreter peers and Deaf community members expressed the importance of examining healthy relationships and challenges when “…everybody knows everyone (Interpreter #15)…” or when personal and professional lives overlap. Regarding the not being seen as equivalent professionals sub-domain, interpreters noted professional hierarchies when working in the medical field.

#### 3.3.3. Physical Health

Likely because of the emphasis in the previous literature [4,5,6,7,8,9,10,11,12,13,14,15,16,18,19], concerns categorized into the physical health domain were the most recognized and well-known contributors to the occupational health of sign language interpreters. However, the reported physical health contributors paled in comparison to mental health and organizational culture (Figure 1; Appendix A). “There was this biofeedback project that I participated in and it was all about signing style and ergonomics. I think those things are important but I also think that we don’t have control over that in a lot of ways (Interpreter #18)”. Physical effects of non-physical work (emotional, cognitive, and linguistic) and aging were identified as sub-domains of physical health. The impact of stress on posture and cognitive processing when interpreting from sign language to English or another spoken language was noted to contribute to the physical effect of interpreting. Similar to how limited extralinguistic knowledge was attributed to increased stress in the mental health domain, a lack of language fluidity also contributed to physical health effects. Many interpreters expressed a decreased tolerance to the physical strain of interpreting with age and were concerned about being able to sustain, “We don’t think about ourselves first but we need to because 25 years ago I didn’t think about these things. Twenty-five years in, I am thinking about these things because they’re starting to affect me at my age. I’m starting to see those differences that had I not been focused on working out over the past, quite a few years, I would not be where I am still because physically you cannot keep doing it (Interpreter #22)”.

There were also some differences in physical exposures noted across settings; specifically, with the time demands and lack of transition or ability to decompress in between assignments when interpreting remotely versus in person and for Deaf interpreters when working with Deaf-Blind community members.

#### 3.3.4. Other Important Occupational Safety, Health, and Well-Being Concerns

Other important occupational safety, health, and well-being concerns included being able to dialogue about the work (debrief) with one another, as well as interpreters being able to safely describe their rationale for ethical decisions and offer/receive feedback (Figure 1; Appendix A). In the last domain of work–life interference, interpreters also expressed that their mental health from responsibilities outside of work (e.g., caregiving for a loved one) compounded expectations on the job or vice versa, that their mental health from interpreting compounded efforts to care for their children.

### 3.4. Solutions for Management of the Occupational Safety, Health, and Well-Being Concerns Reported by Sign Language Interpreters

“I would say within our profession there’s not a lot of talk about it [occupational health] or emphasis on it and it’s only when we wind up hurting ourselves and we start looking for solutions (Interpreter #19)”. Interpreters shared strategies, such as supervision, being present, establishing boundaries, and reframing for management of their reported occupational safety, health, and well-being concerns (Table 2).

#### 3.4.1. Solutions for Management of Mental Health

The selection of interpreting assignments, selection of teams, having economic sense, and prioritization of jobs were common strategies for maintaining occupational safety, health, and well-being. “When I went into private practice, I was trying to be savvy enough to have some balance in my assignments, like the medical work was very emotionally taxing but I loved doing it. The college work was very physically taxing [signed fast pace of the speaker] but intellectually challenging, loved that. I did theater work and that was relax, have fun, and let’s do some art… that was definitely part of the way I did things [signed balance] (Interpreter #16)”.

The expectation that interpreters seek mentorship and having funds available from state-level commissions to pay interpreters for serving as mentors was another successful solution for management. Finding brief diversions, destressing, or decompressing was felt to be important: “If I’m onsite doing an interpreting assignment and I’m driving home, I have that time between work and home… I listen to a Podcast, totally different, change gears, hopefully it is something that is light or whatever. And then I get home and I’m ready to come in. Before I did that, I’d come home and I’d be just stressed and you know, that stacking thing? You’ve got 20 different things that went hard that day and then something minor happens at home and you blow up and you’re like, why? That wasn’t even a big deal but I’m mad or I’m emotional and I don’t even know why, so that destressing or decompression time is helpful (Interpreter #23)”.

Becoming an activist for the interpreting community with state licensing boards and getting involved with establishing educational programs for reducing the experience of secondary traumatic stress or vicarious trauma when witnessing systemic oppression of the Deaf community by the police were additional shared strategies for managing mental health. Access to an employee assistance program was mentioned but interpreters expressed concerns regarding privacy and barriers with eligibility criteria. Pre- and post-conferencing with other practitioners and relying on gate keepers (colleagues and Deaf community members that interpreters can safely approach and authentically dialogue with about work questions or concerns) were just a few of the other commonly shared solutions.

#### 3.4.2. Solutions for Management of Mental Health and Organizational Culture

Because of the lack of work–life balance or boundaries, the loss of agency or loss of self, and deprioritization of self across mental health and organizational culture domains, respectively, interpreters shared that finding their passion outside of work or finding their voice was critical, “If I just stopped interpreting, who would I be? I don’t even know and that’s when I started to branch out and do some other things to, I guess, regain my sanity. I don’t know how common that is, I don’t know if anybody else has been in that situation but for me I think it was really important to actively foster interests in other areas and have a life outside of interpreting (Interpreter #20)”.

Truth telling or being truthful with oneself about the state of affairs within the field and creating safe spaces/communities of supported practice were other identified solutions for management of both mental health and organizational culture.

#### 3.4.3. Solutions for Management of Mental and Physical Health

In addition to managing physical health concerns, exercise was a strategy used for management of mental health, “I made it a priority to work out and I work out at Crossfit. Just two nights ago, I was doing 50 bench presses with 50 pounds. That is not only emotionally releasing, but it also strengthens what I need to keep doing the job that I do (Interpreter #22)”.

#### 3.4.4. Solutions for Management of Physical Health

Ergonomics/posture, workstation set-up, and biomechanics were all shared as solutions for management and prevention of physical health concerns.

## 4. Discussion

The intention of this work was to guide the adaptation of a previously developed TWH program [39,40] by generating a summary of the reported occupational safety, health, and well-being concerns and shared solutions for management from sign language interpreters across settings. Aspiring to purposive maximum variation sampling, unstructured focus groups were conducted with hearing sign language interpreters across regions of the United States, as well as internationally, and with Deaf sign language interpreters. Overall, findings revealed that the paramount concerns of interpreters related to their mental health, such as workplace violence, secondary traumatic stress or vicarious trauma, lack of work–life integration or boundaries, and loss of agency or loss of self, as well as concerns with organizational culture, which seemed to foster deprioritization of self, oppression, elitism, and sexism (Figure 1; Appendix A). One interpreter expressed, “…to not improve says we don’t need to improve and to not examine is like, “yea, everything is great,” but the only way forward is by rectifying and adjusting in a more healthy way (Interpreter #12)”. For the domains and sub-domains of occupational safety, health, and well-being concerns mentioned, there were a greater number of solutions offered for management (Table 2). These included but were not limited to prioritization of jobs, finding autonomy/job control, brief diversions, establishing a support system, creating safe spaces/communities of supported practice, exercise, and ergonomics/posture.

This study centered on the psychosocial environment and the interplay between workplace violence [31], organizational culture, emotional or psychological safety [32], and secondary traumatic stress or vicarious trauma as occupational safety, health, and well-being concerns of sign language interpreters. The National Institute for Occupational Safety and Health at the Centers for Disease Control and Prevention identifies four different types of workplace violence. Type I occurs less frequently and is considered to be of criminal intent, like trespassing or robbery. The most common is Type II or consumer-on-worker violence. Type III, which is also referred to as lateral or horizontal violence, includes bullying among coworkers, and Type IV is when violence about a personal relationship occurs within the work environment [49]. Some previous research specific to the field of sign language interpreting has investigated the prevalence of Type III workplace violence [31]. Ott (2012) [31] described horizontal violence as “infighting within a group of people who experience stress-related to powerlessness,” including behaviors intended to cause harm, stress, and anxiety, like gossiping, criticism, diminishing comments, rudeness, and devaluating others’ professional worth. “… the way that we can eat our young or not support each other (Interpreter #12)” was a comment from one interpreter in this study that was categorized into the organizational culture domain. Another interpreter in this study had surveyed other interpreters who shared their experiences or observations of both Type II and Type III workplace violence, “a majority of respondents said that they had either experienced those things [behaviors associated with horizontal violence, like scapegoating, breaking of confidences, gossip] themselves from other interpreters or from Deaf community members or they had witnessed those things happening in their community (Interpreter #9)”.

Previously identified causes of horizontal violence, such as stress from oppression, stress from subjugated professional status, stress from constrained decision latitude, stress from professional hierarchies, and role stress [31], were echoed in this study and supported by other past evidence [11,30,50]. Because those who have experienced oppression tend to perpetuate oppressive behavior, horizontal violence has been shown to be common in predominantly female service professions, like interpreting, nursing, and education [31]. As mentioned, the field of sign language interpreting reflects between 75 and 82% female [34,46,47,48]. In one of this study’s Deaf interpreter focus groups, it was shared that Deaf interpreters tend to bully each other more than hearing interpreters. When asked why, it was explained, “A Deaf person’s upbringing, being deflated by a hearing teacher saying, “that’s not right”. It’s hard for Deaf interpreters to separate from the oppression they experienced growing up upon entering into the interpreting profession. Sometimes, it’s an uneasy feeling when you see a Deaf interpreter being so critical (Interpreter #24)”.

Characteristics of oppressed groups like low self-esteem [31] was noted by the following sentiment in the deprioritization of self sub-domain of this study, “…self-worth, like viewing yourself as a person who is worthy of care and love that is not just this machine that is here to serve and take care of other people (Interpreter #21)”. Harvey (2019) [30] noted that regularly witnessing oppression can cause interpreters to behave like oppressed groups, which was also shared in this study, “…seeing that the system is not set-up for Deaf people… like the systems that we’re working within… those things that are way out of the control of what we can do as interpreters but that we see every single day, I think it contributes to the strain of the job (Interpreter #21)…”

Stress from subjugated professional status was noted by an interpreter in this study who commented in the workplace violence sub-domain about vertical violence, rather than horizontal violence, due to the system of power when serving on a Deaf-hearing interpreter team. Subjugated professional status was also noted in the oppression sub-domain by an interpreter who had surveyed Deaf interpreters about their experiences working with hearing interpreters and hearing interpreting agencies, “100% of the participants said that they still experience oppression on the job from the hearing team and around 60-something percent experience agency oppression (Interpreter #25)”. Interpreters have also been found to have low decision latitude, which combines skill discretion or the ability to utilize resources along with decision authority or the ability to implement such resources [11]. Stress from professional hierarchies was noted when one interpreter stated, “I think there is a lack of education for people who we work with, especially in a medical field, that don’t see us as equivalent professionals (Interpreter #1)”. Park et al. (2017) [50] found that medical interpreters experienced role challenges when making cultural adaptations or translating cultures and with maintaining professionalism and accuracy. The loss of agency or loss of self, deprioritization of self, and unhealthy relationships with interpreter peers and community members found in this work all support role stress as another contributor to horizontal violence.

Other previous research specific to the field of sign language interpreting has investigated the lack of emotional or psychological safety [32]. The findings of this study were in support of the described experiences in Hill (2018) when interpreters felt unsafe while working or teaming with another interpreter. Psychological safety when teaming involved interpersonal trust, mutual respect, and a comfort-level with being able to make corrections to the interpretation [32]. Interpreters in the current study expressed the selection of teams as an important solution for management, likely, relating to their emotional or psychological safety concerns, “I’m going to be honest. I’ve stopped working with teams I don’t know… it limits my workability, limits which jobs I’ll take but I can’t on the VRI [video remote interpreting] situation now. I will only work with teams that I know (Interpreter #17)”.

Microaggressions, as a subcategory to psychological safety when teaming, were reportedly committed against Deaf interpreters, interpreters of color, Deaf-parented interpreters, and interpreters, in general, because of their appearance [32]. In the mental health domain of this study, there was support for this when two interpreters in the same focus group expressed, “BIPOC interpreters talk about how they feel more stress that impacts their overall health. Deaf interpreters, as well, feel stress. We need to do follow-up research to focus on Deaf interpreters, BIPOC, or black and brown interpreters. I’m sure it is a lot higher than white interpreters (Interpreter #25). The same for LGBTQ interpreters, rainbow interpreters, as well (Interpreter #24)”. Additional support for microaggressions was found in the workplace violence sub-domain when one interpreter shared, “upon presenting to a person who will identify me as white, fem, neurotypical, they will interact with me in one way and then, when I self-disclose my identities, they will interact with me in a different way (Interpreter #9)”. Just like stress from constrained decision latitude was one of the previously identified causes of horizontal violence [31], Hill (2018) also noted that limited decision-making control either by the team interpreter, the interpreter’s skillset, or the interpreting setting (specifically, VRS interpreting) related to safe and unsafe interpreting. This was exemplified by a comment from one interpreter in the setting-specific sub-domain of mental health who was unable to request a remote team when processing a 9-1-1 call because doing so would constrict the Deaf person’s view of the interpreter, “… for that reason, I choose not to team and sometimes it’s at my own expense (Interpreter #8)”. This lack of safety has been shown to hinder interpreter’s performance and contribute to feelings of vulnerability, unworthiness, and shame [32].

Darroch and Dempsey (2016) [29] conducted a systematic review and found that interpreters develop vicarious trauma from continuous exposure to transferential dynamics or transference and countertransference (the mutual impact of the Deaf consumer and sign language interpreter on one another). Comments from interpreters in this study illustrated concerns from continuous exposure on the job, “More so than there being any one particular job that had some negative impact on me, it’s more cumulative (Interpreter #21)” and “…people think it’s AN incident where to me, not to be overly dramatic, it’s like death by a thousand cuts. So, you’re taking it and taking it… you’ve GOT TO have something that counters all of that (Interpreter #16)”. Experiences of secondary traumatic stress or vicarious trauma and workplace violence have been associated with burnout [25,51]. Previous work [23,25,26] has indicated low levels of burnout, on average, in sign language interpreters. However, upon looking at frequencies rather than means, there is some evidence of more moderate to high burnout levels [23,24]. Like Ott (2012) [31] noted for the workplace violence, and Hill (2018) [32] noted for safe and unsafe interpreting, Schwenke (2012) [24] also noted that constrained decision latitude was associated with burnout. Here, we see that the utilization of autonomy/job control, as a solution for management, can serve as a protective factor against burnout. In this study, an interpreter shared that utilization of autonomy/job control was also a solution for management of vicarious trauma, “I think also what my understanding of the best way to mitigate, if not resolve vicarious trauma is to feel as though you have some autonomy while you are in the moment (Interpreter #10)”.

Before our study’s findings can be used for adapting a previously developed TWH program [39,40], it is important to take into account its limitations and methodological strengths. The sample size would be considered small and thus has limited generalizability in a quantitative sense. However, the qualitative methodology used along with the representativeness of interpreters across settings and regions of the United States, as well as internationally, allowed for some theoretical generalizability. The solutions for management (Table 2) were gathered from interpreters’ work-related experiences and are a positive contribution to the literature. While they offer effective strategies for the individual worker, they only include a few suggestions for impacting meaningful and sustainable change at an organizational level. Individual-level factors will only foster occupational safety, health, and well-being to a limited extent. Organizational culture of the field indicated that addressing organizational-level, along with individual-level factors, will be important for future work.

## 5. Conclusions

This work emphasizes the need for increased attention to the mental health concerns of sign language interpreters and the need for transactional (policies and programs) and transformational (leadership and organizational culture) change. May it offer some validation to interpreters who may be struggling. Protecting and promoting the occupational safety, health, and well-being of interpreters must address the psychosocial environment and the interplay between workplace violence, organizational culture, emotional or psychological safety, and secondary traumatic stress or vicarious trauma. Recognizing mental and physical health concerns and individual- and organizational-level factors, this study contributes to the evidence and supports the evolution of future research, practice, policy, and capacity building, aspiring to a more integrated approach.

## Figures and Tables

**Figure 1 ijerph-21-01400-f001:**
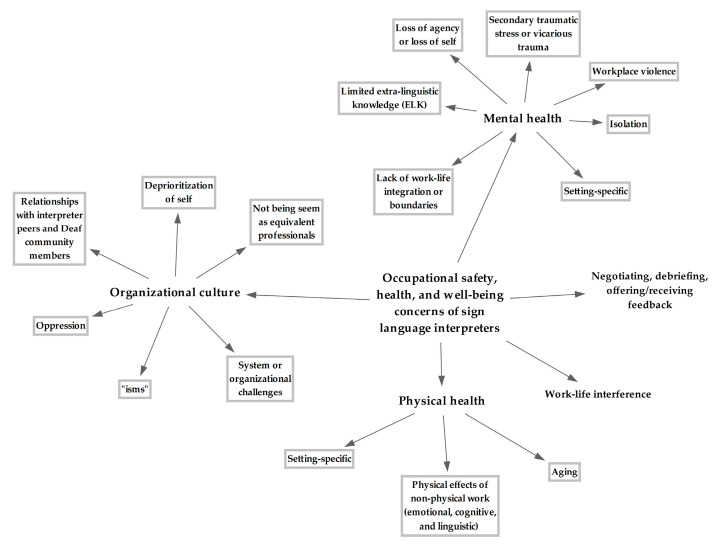
Occupational safety, health, and well-being concerns of sign language interpreters.

**Table 1 ijerph-21-01400-t001:** Participant demographics.

	Total Sample (*n* = 27)
Age (mean ± SD)	53.7 ± 10.7
Years of interpreting experience (mean ± SD)	27.0 ± 10.7
Interpreting hours/week (mean ± SD)	21.3 ± 13.5
Primary interpreting setting (*n*, %)	
Community freelance	8 (29.6)
Government	2 (7.4)
K-12 education	2 (7.4)
Legal	2 (7.4)
Medical	3 (11.1)
Postsecondary education	3 (11.1)
Video relay	3 (11.1)
Video remote	4 (14.8)
Interpreting certification * (*n*, %)	
Certified Deaf Interpreter (CDI)	4 (14.8)
Certificate of Interpretation/Certificate of Transliteration (CI & CT)	11 (40.7)
Educational Certificate: K-12 (Ed:K-12)	2 (7.4)
Educational Interpreter Performance Assessment (≥4.0; EPIA)	5 (18.5)
National Association for the Deaf-Generalist (NAD III)	2 (7.4)
National Association for the Deaf-Master (NAD V)	1 (3.7)
National Interpreter Certification (NIC)	13 (48.2)
National Interpreter Certification-Advanced (NIC-A)	1 (3.7)
Specialty Certificate-Legal (SC:L)	1 (3.7)
Transliteration Certificate (TC)	1 (3.7)
Hearing status (*n*, %)	
Deaf	4 (14.8)
Hard-of-hearing	1 (3.7)
Hearing	22 (81.5)
Gender (*n*, %)	
Female	22 (81.5)
Male	2 (7.4)
Genderqueer	1 (3.7)
Non-binary	2 (7.4)
Race/ethnicity (*n*, %)	
Asian	0 (0)
Black or African American	1 (3.7)
Hispanic or Latino	1 (3.7)
Multiracial	2 (7.4)
White	23 (85.2)
Geographical region ^‡^ (*n*, %)	
Central (Colorado, Iowa, Texas)	4 (14.8)
Midwest (Indiana, Michigan, Minnesota, Wisconsin)	4 (14.8)
Northeast (Massachusetts, New Hampshire, New Jersey, New York)	6 (22.2)
Pacific (Arizona, California, Oregon, Utah, Washington)	5 (18.5)
Southeast (Florida, North Carolina, Tennessee, Virginia)	4 (14.8)
Internationally-based	4 (14.8)
Highest level of education (*n*, %)	
Less than a high school diploma	1 (3.7)
High school diploma or equivalent	0 (0)
Associate degree	3 (11.1)
Bachelor degree	11 (40.7)
Master degree	9 (33.3)
Doctorate degree	3 (11.1)

Notes: SD = standard deviation; K-12 = kindergarten through twelfth grade; * = some participants maintain more than one interpreting certification; ^‡^ = more than one interpreter was represented from Iowa, New Hampshire, and New York.

**Table 2 ijerph-21-01400-t002:** Solutions for management of the concerns reported by sign language interpreters.

Occupational Safety, Health, and Well-Being Concerns	Solution	Exemplar Quote
Mental health	Selection of assignments, selection of teams, economic sense, and prioritization of jobs	The reason why I take postsecondary freelance work is to avoid things like vicarious trauma. That’s the reason I only lasted three years doing VRI/VRS interpreting is because it’s just too much to be that involved with people’s lives for me. The stress that they carry and the phone calls that they’re making had an impact on my life and my work. With postsecondary interpreting, the roles are pretty clear. This person is teaching, this person is learning (Interpreter #14).
		It is a strategy in a backwards way around well-being. This is my goal for the month, I need ‘x’ amount to pay my bills, to keep the house going, and then, the rest of it is quote/unquote cream. And then, overdoing it for months I knew we were going to be low on work, like the summer months… just having some economic sense is actually part of taking care of yourself (Interpreter #16).
		I’ve made a choice to make less money to only work in situations that I am confident. I’ve turned down work where I’m not putting myself into that. I’m not doing it. I’ll lose money over it but it’s too much on my body, too much on my mind to actually engage with that. I’m going to be honest, I’ve stopped working with teams I don’t know… it limits my workability, limits which jobs I’ll take but I can’t on the VRI situation now. I will only work with teams that I know. I have the privilege to be able to do that, like that means that all of the work I’m not taking because I’m making a privileged choice is hard to deal with emotionally, like the amount of work that I’m choosing not to take, someone needs to do it. Are they then going to be put into a situation which isn’t as supportive as the ones that I am choosing to limit myself to (Interpreter #17)?
		If I [agency owner] get a phone call from the emergency room and there’s someone who’s been in a bad accident, I can’t say, “sorry, I don’t feel like coming”. I will go. The only way to keep myself with not having an occupational health problem is to say, “theater, you would like to have me come and interpret a play, I can’t do that. Do you have Deaf people coming?” “Oh no, we just want access”. “Well, that’s lovely and I’ve never done this in 25 years but I’m doing it now. Do you have Deaf attendees? Of course, we will make a way to do it. Do you not have Deaf attendees? It’s wonderful that you want to provide access but I don’t have the time or the people right now”. In my world, that emergency room person is top priority (Interpreter #22).
		It’s important, upon receiving a job request, to read about the job before accepting it. Some interpreters just accept, responding quickly that yes, I’m available and go ahead because they need the money without realizing that when they report to the assignment they don’t know anything about what’s going on and subsequently, setting-up the risk to themselves (Interpreter #24).
	Mentorship	There is this emotional component of things that you’re not dealing with but are coming through your hands and things that you experience, like the vicarious trauma that happens when there is a lack of mentorship. Even if it’s not, I’m trying to become a certified interpreter and I need a mentor to help me get ready for the test, but I need a mentor to kindof to show me how you stay afloat through all of this (Interpreter #12).
		We bring in new people and they have to do mentoring before they can start working. It’s a given that we’re doing mentoring hours. Mentoring is paid for us through the [state] Commission for the Deaf and Hard of Hearing. They have a funding source for all of the legal interpreting and are also involved in certifying or training to become certified as legal interpreters, so they do all of that (Interpreter #23).
	Brief diversions, destressing, or decompressing	Sometimes when work is getting really busy, the calls keep coming in super fast, you never know what type of call you’re getting, I’ll do something different. Like, puzzles online. Sometimes I’ll read a story to take me away from a place. That will help me to… something totally different… it helps your mind detach from that stuff (Interpreter #8).
		You have to schedule time for intentional decompression. We don’t realize that we have to do it and then, the time just slips away and you don’t… when you’re in the car, it organically gives you that opportunity to think through or listen to some music or a podcast or just look at the world around you. Maybe that having to be intentional about it makes it easier to forget to do it or easier not to do it (Interpreter #21).
	Autonomy/job control	What my understanding of the best way to mitigate, if not resolve vicarious trauma is to feel as though you have some autonomy while you are in the moment (Interpreter #10).
		I think one of the reasons why I love my job, as passionately as I love my job today as much as or more than I did when I first started my job, years and years and years ago, is that I have complete freedom of choice. I do not have to go to work. I do not have to do this job any more than the one time I’ve agreed to do it. I have the ability to say, this is an oppressive environment, I don’t have to go back. This is an oppressive Deaf person, they are not fun to work with, I don’t have to go back (Interpreter #13).
		We do have control even where we think we don’t have control (Interpreter #14).
		Wherever interpreters can have some autonomy and some self-selection, I think would be helpful because there’s so much time that we spend where we don’t have a lot of presence as just who we are (Interpreter #21).
	Supervision	I have thought about trying to go through the process of becoming a supervision leader because I see that need for it, but I also don’t know how receptive colleagues and others might be (Interpreter #3).
		I’ve been participating in supervision sessions, which has actually been super beneficial to my mental health and well-being… being able to discuss and talk about the work is a really productive way (Interpreter #4).
	Debriefing	I have several friends, in the interpreting community, that I can go to, debrief, and say, “errr, I just had this guy and I was trying to ask for clarification and he said bad interpreter [signed hang up]”. Hang up on me, why? I think just putting it out there and “you’re still a great interpreter [signed double thumbs up]”. Okay I’m good now. Just having the comradery, that feedback, that support, whether it’s just [signed thumbs up], you’re doing a good job… those things help me (Interpreter #8).
		I need to be able to come home and tell somebody who understands why I, as someone who is supposed to be ethical and socially just and like meet all of these very high humanitarian mandates in my profession, had to make a choice between a bad choice and a worse choice today (Interpreter #9).
		The job is lonely. I have to find ways to support myself through difficult assignments or interactions. Because I’ve lasted so long at doing this, I must be able to do that well enough that I haven’t had a crisis of not wanting to continue as an interpreter. But it’s not a given. I definitely have to have the space to remind myself what was hard, remind myself that I can debrief, I should debrief, and talk with trusted colleagues (Interpreter #11).
		Some of those things, you have to be able to digest with someone else and be like, “am I looking at this weird? Or, can I just tell you what happened today?” I’ve gotta get this out and then, putting it out there is like okay, I feel better. I don’t have to wear this. That emotional stress affects your physical health (Interpreter #12).
	Negotiating	When I get a request for an interpreting assignment, I will respond with no, if it doesn’t work at that time, but I could do this and I’ll give them other times. For whatever reason in our field, we were taught not to do that because we don’t want the interpreter’s schedule to dictate that Deaf person’s schedule but the clients seem very receptive to it (Interpreter #21).
	Spirituality	When I have a particularly hard week, I turn to my faith. I listen to uplifting message music in the car and I decompress that way (Interpreter #22).
		For me, my relationship with God helps me so much. God helps me to make what happens smooth. I surrender over to God. It could be interacting with family, at work, and with friends, I use the same strategies to protect myself so that I can carry on (Interpreter #26).
	Being present	What I’ve done for myself, in order to stay away from that feeling of overwhelm, is to stay only in that moment. I had this happen a week and a half ago. A lady who is passing away in front of me. There’s all this noise back here, there’s all these people… she’s the most important person in my world at that moment and I don’t care about the rest. I’m staying in the moment for right here, right now with this lady, so the last thing she sees before her eyes close is someone signing (Interpreter #22).
		When I stress out over those things that I used to stress about at a younger age and go, “oh, I don’t know how you do this”. There’s become more of a quiet to it. If that makes sense? My experiences now are more quiet and they’re more, “wow, did I really learn something from that”. That’s an experience that I never want to forget because it was so beautiful. Whether it was a baby being born or someone passing and everything in between. And that doesn’t mean that we don’t cry and we don’t feel it. I get in my car and I cry all the time because I love the people I work with, but that inner peace that comes from being quiet over the situation because you were in the moment, changes everything. Instead of leaving and going, “oh my goodness and I’ve got to drive home and it’s crazy and I’m gonna get home and I’ve got all these things to do,” I stay in that quietness, and it is very healing (Interpreter #22).
	Establishing boundaries	It takes training, in the beginning when you’re young it’s hard to manage, but with training, you can become more protective of yourself and carry on. To keep at a slight distance from this or that, you have to take care of yourself (Interpreter #26).
		Sometimes hearing interpreters will share [their mental health problems] with me. If it goes on, sometimes I’ll give them tips. If they don’t take my advice, I am direct with them. I let them know that I don’t want to continue talking about that because if I keep listening and I will take on some of their toxicity. I feel like I can’t offer any more, I establish boundaries (Interpreter #27).
		Luckily, I have many hobbies that also serve as my therapy and my escape. I have many outlets. If something sticks to me while at work, when I arrive home, I’m able to let it go before going into the house. I had children that I needed to take care of. I learned how to cope and not let things from work stay with me. I have several positive outlets that help to fill me up when my tank is low. If something becomes a problem at work, I feel sorry for that person, I give them my best for communication access, I work hard and make sure everything comes together for that consumer, but to bring it home with me, I don’t do that (Interpreter #27).
	Become less involved	What I have found to protect myself, because of the stress from VRS interpreting, is that I don’t go to the Deaf community much at all, which saddens me but at the same time it’s about the only way I can protect myself from feeling so overwhelmed with that need to advocate at the same time I interpret (Interpreter #7).
	Support system	I find a support system outside of the field of interpreting and outside of Deaf people. I have a small group of friends and family members that if I am either going someplace that I feel very unsafe… I live in the South, I’m black. There are times that I am sent to places that it really is not safe for me and I’ll go cause I’m not scared, but I will alert other people. I do that and that gives me internal peace that I am just not out there in the wild world without anybody else being aware (Interpreter #6).
	Reframing	There are other interpreters who are highly skilled and can be put in the same situation that would put my hair on fire and it has absolutely no impact on them at all. Despite my pessimism, I try to go there… they’re going to get somebody who is a better fit than I am, someone who is qualified for this particular topic, situation, whatever the case may be, and recognize that I’m not the perfect interpreter for everybody (Interpreter #19).
	Activism	I have kept abreast of some unfortunate short-sided decisions that the [interpreting] licensure board has made. I actively engage in sending letters to say, “hey, you need to think about the ramifications of your decision,” knowing it’s not going to go anywhere, but still, at least I know that they were informed. Those are ways, so that I don’t get this feeling of helplessness (Interpreter #6).
	Educational programs that decrease the oppression of the Deaf community	I’m working with a state patrol police type of person who used to be in probation to develop an education program for people that work in the prisons and the police department about communication, what is ADA, and all of the things that we wish that they would know. We were working on it for a couple of years and we’ve rolled it out a little bit. We’re not finished yet, but I can tell that the incarcerated facilities where we have gone (prisons, country jails, courts, etc.) and we’ve gone more than one time, I see a lot less problems and complaints, let’s say complaints, from people who use sign language. It is having an impact but it’s a matter of taking people through it and the agencies will say, well I can only afford to take five people off the floor today because we are also short on staff, so it’s slow to roll out (Interpreter #23).
		The County Sheriff Department is doing the training… how to interact with people who use sign language in more safe and respectful ways. The person who is heading the program, I’m helping her, but the person who heading the program is a certified sign language interpreter as well as a sheriff and has been in the parole, as a parole officer, so has a lot of experience in various areas of that whole community, has a lot of contacts with the people in the agencies and able to get in in ways that nobody else probably can get in. They saw what we all see that’s not fair, how can we fix it, we can’t, we’re just the interpreter. But being not the interpreter, being someone else, they have the way to go ahead and get rid of some of those barriers (Interpreter #23).
	Employee assistance program	There’s an employee assistance program, but it’s very limited, like you can call up to this many times. You can’t guarantee that you’re going to talk to the same person and to be perfectly frank, I don’t know that any of us really trusts that what we are really talking about with whomever we are talking to, like how is this being reported? Is it being reported that we did use it? Is it being reported that, hey, you need… I don’t know that my rights under HIPAA apply to employee assistance. I don’t know who these people are. I certainly didn’t get anything from them in writing saying here are your privacy rights. Even though that benefit is there for us, nobody uses it (Interpreter #3).
		I do believe the corporation is trying to provide individual resources, like the EAP counseling if you have a really stressful time; however, there are a lot of hoops to jump through if you want to get those resources and, especially, if you are part-time. What I end up doing is just finding ways that I can lessen my load because trying to advocate for carrying the load efficiently, for me, has not been working (Interpreter #7).
	Pre- and post-conferencing with other practitioners	While we were interpreting, one of the patients had a failure and they were going to be kicked out of the program. After the whole thing was done, one of the interpreters working that evening basically stood outside in shock and one of the therapists came up and was like, “are you okay?” She [the interpreter] went into some stuff and because of that interaction, interpreters were invited to the pre- and the post-conferences with the therapists and the therapists were transparent with the patients by saying, we will be meeting with the interpreters so they know what our plans for the evening are, we will be debriefing with them so we can get language information, as well as making sure everyone on the team is having the same understanding, which was incredible (Interpreter #2).
	Gate keepers	I always feel like there are interpreting colleagues that I turn to and turn to me and Deaf community members that I turn to and turn to me. That gate keeper concept [a group of people who are either hand selected or taken under the wing of “x” Deaf person or “x” Deaf family. They are your cheerleader, they’re your strongest critic, they are all of that for you] is very much alive in my practice and it’s not as easy for the younger generation to generate that. You have to have colleagues and community members you can sit down with and quote/unquote say it wrong so that you can get some authentic discussion going on. I take so for granted that exists in my life because I know it doesn’t for many people (Interpreter #16).
Mental health and organizational culture	Finding your passion outside of work/finding your own voice	[when talking about enjoying the outdoors, biking, and taking up photography] Get out there and find your passion outside of work, something that makes you feel alive (Interpreter #8).
		It was just finding the things that were missing and making active choices rather than passively just accepting, like capitalism and I need to make money to live. It’s not easy. Money is a factor here and it’s not something we can ignore. People do need to earn money to live, so you know there’s a balance. But looking actively at what was keeping me from my whole self (Interpreter #17).
		It’s like being able to use your own voice for your own self and I think that is important to be able to have something in your life where you’re able to express your own voice (Interpreter #18).
		That’s interesting talking about not being able to be yourself because the one thing I’ve really enjoyed about what I’m doing in another professional realm is that I CAN be myself. I can say whatever I want whenever I want. I’m not sitting there relaying a message. I have a lot more space to just be who I am (Interpreter #20).
	Truth-telling	The other thing that I have done is this idea of truth telling. I tell myself the truth about the state of affairs in this field. I have intentionally been an observer of trends and allow the evidence to guide me versus thinking about what ideally should be (Interpreter #6).
	Creating safe spaces/communities of supported practice	This [referencing the focus group] is good and healthy. Shame dies where stories are told in safe spaces and interpreters don’t have safe spaces to normalize their experiences with each other and that is a problem (Interpreter #9).
Mental and physical health	Exercise	I swim almost every day. I use the exercise time, the me time, the unplug from everything, just look at the bottom of the pool… I mean, I’m swimming laps, umm… to process a lot of that [difficult assignments and interactions], so it is related for me that the physical and more emotional aspects are not discrete from each other, they’re very related in my body (Interpreter #11).
		I know how to take care of myself. I work out. I go to the gym. I go and do things I really enjoy, it helps my mind detach. I set rules for myself. I don’t bring problems to work (Interpreter #27).
Physical health	Ergonomics/posture	We had a certified Deaf interpreter who works with novice interpreters come in and do an analysis, not only of your sign production and your work, but he also gave feedback, “do you realize that you’re kicking your hip out when you’re standing there, which leads to your spine being off kilter?” …noting those types of physical things (Interpreter #2).
		I just think the newer interpreters are going to need that ergonomic stuff because they’ll hurt themselves. I was a perfect example (Interpreter #16).
	Workstation set-up	Ergonomic evaluators would come in and see how you sit and teach you how to adjust your chair and what to do with the armrests on your chair, so that was very helpful and I do that to this day. I work at home but I set up my home office the same way, so that’s all really good (Interpreter #23).
	Biomechanics	In standing, instead of keeping my hands at my chest, I put them down along the sides of the body because that releases tension in the arms, those sorts of things… (Interpreter #2).

Notes: ADA = Americans with Disabilities Act; EAP = employee assistance programs; VRI = video remote interpreting; VRS = video relay service.

## Data Availability

Requests to access the datasets should be directed to the corresponding author.

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
