# Peer review of "Occupational Safety, Health, and Well-Being Concerns and Solutions for Management Reported by Sign Language Interpreters: A Qualitative Study"

_ijerph, 2024, doi:10.3390/ijerph21111400_

Round 1

Reviewer 1 Report

Comments and Suggestions for Authors

Thank you for inviting me to review this interesting paper on occupational safety, heath and well-being among sign language interpreters.

Here are some suggestions to improve the paper further:

Introduction section

·       The aim could be more clearly described in the last paragraph. In the current version of the manuscript, it is somewhat unspecific and “wordy”, and more a description of what was done, rather than a clear aim.

Materials and Methods

·       Participants were recruited from RID. Please provide more information on number/proportion of members.  

·       The authors refer to another paper (Roman et al 2023b) for sample size estimation. I recommend including a clear description of sampling method, typically used in qualitative research, for example purposeful sampling.

·       I also miss description on number of participants in each focus group and a clear description of the informed consent process.

·       I also miss a description on content in the focus groups. They are described as unstructured. Was no question/topic guide used?  What strategies were used to involve all participants and to encourage dialogue between participants?

·       Four participants are described to come from outside the US. The authors explain that no information is provided regarding origin, due to anonymity. I suggest that you provide information on what continent these participants come from.

·       I recommend including a figure describing an overview of the identified domains during the analytic process. The domains are listed in Table 2, but there are also exemplar quotes includes included, resulting in a very large and hard to read table. Table 2 might be better as a supplementary or appendix.

·       Table 1 would be easier to read if the text in the first column were left-justified.

Discussion

·       The outline of the discussion could be improved if each of the domains were discussed in order and in separate paragraphs.

·       I am missing a paragraph on methodological strengths and weaknesses.

Conclusion

·       Well-formulated but I recommend caution on generalizability, especially internationally and no description of sampling method.

Author Response

October 3, 2024

The authors thank the Assistant Editor and Reviewers for the time and effort taken to give our manuscript entitled, “Occupational safety, health, and well-being concerns and solutions for management reported by sign language interpreters: A qualitative study,” a thorough and thoughtful review. Please see our responses detailed below (in red) to each of the raised comments. Thank you for your willingness to evaluate the revised version of our manuscript for further publication consideration in the International Journal of Environmental Research and Public Health.

Reviewer 1:

Thank you for inviting me to review this interesting paper on occupational safety, health, and well-being among sign language interpreters.

Thank you for your positive review of our initial draft and for your suggestions on how to improve its clarity. Our specific responses to your individual comments are detailed below.

Here are some suggestions to improve the paper further:

Introduction section

1) The aim could be more clearly described in the last paragraph. In the current version of the manuscript, it is somewhat unspecific and “wordy”, and more a description of what was done, rather than a clear aim.
We have made edits to the last sentence in the last paragraph of the Introduction. It now explicitly reads, “Our aim was to create an open forum for sign language interpreters to express their occupational safety, health, and well-being concerns, as well as share their solutions for management.”

Materials and Methods

2) Participants were recruited from RID. Please provide more information on number/proportion of members.  

We moved our statement about maximum variation sampling in the Data collection sub-section of the Materials and Methods to the Participant sub-section. We also added that maximum variation sampling was purposive maximum variation sampling and elaborated more specifically about how we contacted leadership across affiliate chapters and member sections of RID. We moved information about the five regions of RID from the Data collection sub-section to the Participant sub-section. The first paragraph of the Introduction reports 10,385 certified members in RID, so the reader should be able to infer the proportion. We also added that we specifically recruited interpreters who identify as male and/or are Deaf interpreters who publicly shared their contact information on RID’s website.

3) The authors refer to another paper (Roman et al 2023b) for sample size estimation. I recommend including a clear description of sampling method, typically used in qualitative research, for example purposeful sampling.

We hope our response to comment 2 above addresses this concern.

4) I also miss description on number of participants in each focus group and a clear description of the informed consent process.

The Data collection sub-section reports that “One focus group occurred across each of the five regions of RID, as well as one with English and American Sign Language interpreters based outside of the United States and two with certified Deaf interpreters.” In the Participants sub-section of the Results, we now report the specific breakdown for each focus group, “Nineteen interpreters participated in the focus groups across the Northeast (n=4), Southeast (n=4), Midwest (n=4), Central (n=3), and Pacific (n=4) regions of RID. Four interpreters participated in the focus group with interpreters based outside of the United States and four interpreters (or two interpreters in each) participated in the two certified Deaf interpreter focus groups.”

In the beginning of the Materials and Methods, we reported, “This qualitative research study (STUDY00008720) underwent review by the University of Rochester’s Research Subjects Review Board and based on federal and University criteria, it was deemed exempt.” This sentence has now been changed to, “This qualitative research study (STUDY00008720) underwent review by the University of Rochester’s Research Subjects Review Board. Based on federal and University criteria, it was deemed exempt and thus, the requirement of written informed consent from participants was waived.” In lines 156-158 of the Data collection sub-section in the Materials and Methods, we describe that study procedures and expectations were explained and participants were given the opportunity to ask any questions. Also, at the end of the article, information within the Institutional Review Board and Informed Consent Statements (page 21, lines 583-588) should address any concerns relating to providing a clear description of the informed consent process.

5) I also miss a description on content in the focus groups. They are described as unstructured. Was no question/topic guide used?  What strategies were used to involve all participants and to encourage dialogue between participants?

Lines 156-162 of the Data collection sub-section in the Materials and Methods contains information about the organization of the focus groups. Upon explaining the study procedures and expectations, participants were encouraged to allow everyone an equal opportunity to contribute. Our goal was to create an open forum. Due to the nature of the focus groups being unstructured, no question guide was used and dialogue evolved organically. We were more specific about this on Page 4 in lines 159-162.

6) Four participants are described to come from outside the US. The authors explain that no information is provided regarding origin, due to anonymity. I suggest that you provide information on what continent these participants come from.

In the Participants sub-section of the Results, we added that interpreters based outside of the United States were from the continents of North America, South America, and Europe.

7) I recommend including a figure describing an overview of the identified domains during the analytic process. The domains are listed in Table 2, but there are also exemplar quotes includes included, resulting in a very large and hard to read table. Table 2 might be better as a supplementary or appendix.

We wanted to be able to honor the contributions of the participants by conveying their exemplar quotes, however we understand and agree that Table 2 is very large. As suggested, we have now provided a Figure (Fig. 1) including the domains and sub-domains from the data analysis and are okay with moving the original Table 2 to supplementary material. We added the following statement at the end of the manuscript, “Supplementary materials: A detailed description of the domains, sub-domains, and exemplar quotes resulting from the rapid qualitative analysis of the focus groups can be found in Table S2: Occupational safety, health, and well-being concerns of sign language interpreters.” We added reference to the new (Figure 1; Supplementary materials: Table S2) where appropriate and updated the number of the former Table 3 to now reflect as Table 2.

8) Table 1 would be easier to read if the text in the first column were left-justified.

Apologies but this comment is unclear, as the first columns of all three tables are left-justified. This might be related to how the tables convey within the manuscript template provided by the journal. Within the template, only the References are fully left-justified, otherwise the entire body of the manuscript is indented.  

Discussion

9) The outline of the discussion could be improved if each of the domains were discussed in order and in separate paragraphs.

After our summary introductory paragraph in the Discussion, we now offer the following topic sentence to better establish expectations for the forthcoming Discussion, “The study centered on the psychosocial environment and the interplay between workplace violence [30], organizational culture, emotional or psychological safety [31], and secondary traumatic stress or vicarious trauma as occupational safety, health, and well-being concerns of sign language interpreters.” Then, we go on to elaborate on how this study’s findings supported the previous evidence on horizontal violence (and the identified causes of), emotional or psychological safety (and microaggressions, as a subcategory of psychological safety when teaming), and secondary traumatic stress or vicarious trauma experienced by interpreters. There is overlap between the mental health and organizational culture domains, so it would be difficult to parse them out separately.

10) I am missing a paragraph on methodological strengths and weaknesses.

The study’s limitations and methodological strengths were embedded in the Conclusions. They have been pulled out and now create the final paragraph of the Discussion.

Conclusion

11) Well-formulated but I recommend caution on generalizability, especially internationally and no description of sampling method.

Thank you. Regarding the concern relating to generalizability, we qualify our statement by saying, “…allowed for some theoretical generalizability…” The descriptor of the purposive maximum variation sampling method was added to the summary introductory paragraph of the Discussion.

Reviewer 2 Report

Comments and Suggestions for Authors

Dear authors, I congratulate the authors on the manuscript, as it contributes to a topic that is of interest and little explored. I only have two comments or questions: 1) the research was not submitted to a research ethics committee? 2) There is no information on the occupational ties of the subjects studied, i.e. whether they are self-employed or employees of a company?

Author Response

October 3, 2024

The authors thank the Assistant Editor and Reviewers for the time and effort taken to give our manuscript entitled, “Occupational safety, health, and well-being concerns and solutions for management reported by sign language interpreters: A qualitative study,” a thorough and thoughtful review. Please see our responses detailed below (in red) to each of the raised comments. Thank you for your willingness to evaluate the revised version of our manuscript for further publication consideration in the International Journal of Environmental Research and Public Health.

Reviewer 2:

Dear authors, I congratulate the authors on the manuscript, as it contributes to a topic that is of interest and little explored.

Thank you for your support of our work.

I only have two comments or questions:

1) the research was not submitted to a research ethics committee?

Thank you for this comment. In the beginning of the Materials and Methods, we report “This qualitative research study (STUDY00008720) underwent review by the University of Rochester’s Research Subjects Review Board. Based on federal and University criteria, it was deemed exempt and thus, the requirement of written informed consent from participants was waived.” At the end of the article, additional information within the Institutional Review Board and Informed Consent Statements (page 21, lines 583-588) should address any further concerns relating to Research Subject Review Board review.

2) There is no information on the occupational ties of the subjects studied, i.e. whether they are self-employed or employees of a company?

The participants shared their primary interpreting setting and the breakdown of representation is conveyed in the Participants demographics table (Table 1). Nearly one third of the sample reported working as community freelance interpreters, which indicates that the interpreters work as independent contractors or are self-employed. We offered this as an example of one of the represented interpreting settings in the Participants sub-section of the Results for readers who may be unfamiliar.

Reviewer 3 Report

Comments and Suggestions for Authors

Thank you for this study, doing it with a qualitative methodology is very interesting to the existing literature.

The country where it is carried out should appear in the title.

On line 59 onwards, mental and psychosocial fatigue should be included as a mental health risk. 

Line 71 I think this should be explained and described better.

Line 117 what is the objective of this study is not clear to me.

Line 123 Why was more than 10 hours of work taken as a reference? Why not at least half a day 20 hours? What were other inclusion and/or exclusion criteria?

In material and methods the COREQ-32 should be included.

Table 1.- Why was it not asked if they were CODA? Why not ask about physical activity or stretching programs or other factors that affect health in this population?

I think that the quotes would be better in table and specify the SLI profile, as it is put in terms of form is somewhat tedious, it is better as they have it in table 2, specifying profile.

Discussion, the limitations of this study and future lines of research should be included.

The study with doi https://doi.org/10.3390/ijerph19159038, studied not only physical health but also mental health, social function and cognitive and psychosocial phatic it would be interesting to discuss these quantitative data with those of this qualitative study.

Conclusions 

Removing line 528 “we hope” is not appropriate in an academic paper. The conclusion should be more reduced and respond only to the objective of the study which, as I commented, is not clear to me.

References

There are more recent studies not included... in studies of these characteristics, they should not cite articles more than 10 years old.

Author Response

October 3, 2024

The authors thank the Assistant Editor and Reviewers for the time and effort taken to give our manuscript entitled, “Occupational safety, health, and well-being concerns and solutions for management reported by sign language interpreters: A qualitative study,” a thorough and thoughtful review. Please see our responses detailed below (in red) to each of the raised comments. Thank you for your willingness to evaluate the revised version of our manuscript for further publication consideration in the International Journal of Environmental Research and Public Health.

Reviewer 3:

Thank you for this study, doing it with a qualitative methodology is very interesting to the existing literature.

Thank you for your affirmation of this work.

1) The country where it is carried out should appear in the title.

While the study was carried out in the United States, it did not only include interpreters based in the United States. Thus, we have opted to keep the country where it was carried out not contained within the title. We did add that it was a qualitative study to the title to adhere to the Standards for Reporting Qualitative Research, which was added in response to comment 7 below.

2) On line 59 onwards, mental and psychosocial fatigue should be included as a mental health risk. 

We added mental and psychosocial fatigue and referenced the suggested publication in comment 11 below.

3) Line 71 I think this should be explained and described better.

Thank you for this suggestion. We introduced these mental health concerns as emerging in the Introduction and described how this study centered on horizontal violence and emotional or psychological safety more so in the Discussion.

4) Line 117 what is the objective of this study is not clear to me.

We have made edits to the last sentence in the last paragraph of the Introduction. It now explicitly reads, “Our aim was to create an open forum for sign language interpreters to express their occupational safety, health, and well-being concerns, as well as share their solutions for management.”

5) Line 123 Why was more than 10 hours of work taken as a reference? Why not at least half a day 20 hours?

Thank you for pointing this out. Because our emphasis was on aspiring for a maximum variation (demographic and geographic) sample, in looking into the comment, we realized that a minimum number of interpreting hours per week or years of interpreting experience were not required for participation in the unstructured focus groups. We have used the ≥10 hours per week as part of our inclusion criteria in past work [34,42,48]. It has worked well for us to capture a broader participant pool because interpreters’ hours seem to fluctuate seasonally. To address any concern that a specific number of hours per week may or may not be enough, we have added the average interpreting hours per week (21.3 ± 13.5) reported by participants in this study to the demographics table (Table 1).   

6) What were other inclusion and/or exclusion criteria?

The main inclusion criteria are conveyed in the first sentence of the Participants sub-section of the Materials and Methods. Participants were required to be adults (≥18 years old) and certified sign language interpreters bilingual in English and American Sign Language.

7) In Materials and Methods the COREQ-32 should be included.

Thank you for this suggestion. We have referenced the Standards for Reporting Qualitative Research (SRQR) in the Materials and Methods and included Table S1: Standards for Reporting Qualitative Research (SRQR) in Supplementary materials.

8) Table 1.- Why was it not asked if they were CODA? Why not ask about physical activity or stretching programs or other factors that affect health in this population?

We did not specifically ask if participants were children of Deaf adults (CODAs). We understand that CODAs are, likely, native signers and are aware of their decreased prevalence of reported pain while signing when compared with non-native signers, but that specific characteristic was not particularly sought after for purposes of our intent to create an open forum for sign language interpreters to express their occupational safety, health, and well-being concerns. Regarding asking about physical activity or stretching programs, we did not adhere to a question guide for purposes of these unstructured focus groups. Interpreters were invited to organically share their concerns and dialogue about their solutions for management. In Table 2, interpreters shared that exercise was a strategy utilized for managing both mental and physical health concerns but, overall, concerns surrounding physical health were not as paramount as mental health and organizational culture.

9) I think that the quotes would be better in table and specify the SLI profile, as it is put in terms of form is somewhat tedious, it is better as they have it in table 2, specifying profile.

We did consider having the labels for the exemplar quotes reflect the interpreters’ state within the United States or country. For example, (Wisconsin interpreter #1). Out of our desire for preserving the anonymity of the participants’ contributions, we decided to use the generic label of (Interpreter #1). Because this study provided a summary of concerns and solutions for management across interpreting settings, we also did not think it was important to differentiate the interpreters by labeling the exemplar quotes as (Community/freelance interpreter #1) or (Educational interpreter #3). We think Table 1 does a good job of providing the reader with a sense of the representation across the entire sample. As stated on page 4, lines 188-191, interpreters represented a wide distribution of interpreting settings, interpreting certifications, and levels of education.

10) Discussion, the limitations of this study and future lines of research should be included.

The study’s limitations and methodological strengths were embedded in the Conclusions. They have been pulled out and now create the final paragraph of the Discussion. We added that addressing organizational-level, along with individual-level factors will be important for future work.

11) The study with doi https://doi.org/10.3390/ijerph19159038, studied not only physical health but also mental health, social function and cognitive and psychosocial phatic it would be interesting to discuss these quantitative data with those of this qualitative study.

Upon addressing comment 2 above, we added this suggested study as a citation when presenting the growing empirical work about the mental health concerns of sign language interpreters. We centered our Discussion on how the findings from this study further reiterated the interplay between workplace violence, organizational culture, emotional or psychological safety, and secondary traumatic stress or vicarious trauma. We appreciate you drawing our attention this reference. However, the predominant findings from our study did not particularly reiterate the mental and psychosocial fatigue, social function, and cognitive and psychosocial phatic reported in the suggested study. 

Conclusions 

12) Removing line 528 “we hope” is not appropriate in an academic paper.

We changed “We hope it offers…” to “May it offer…”

13) The conclusion should be more reduced and respond only to the objective of the study which, as I commented, is not clear to me.

We made the study’s aim more evident when responding to comment 4 above. With the movement of the study’s limitations and methodological strengths to the final paragraph of the Discussion, the Conclusions are now more succinct. The objective of the study was repeated in our summary introductory paragraph in the Discussion and thus, is not repeated again in the Conclusions.

References

14) There are more recent studies not included... in studies of these characteristics, they should not cite articles more than 10 years old.

Thank you for this comment. Sharing a lot of the early work investigating the occupational health of interpreters helps to demonstrate that much of the previous literature, dating back to the work of Gary Sanderson in 1987, has traditionally focused on their physical health. There was very little previous research regarding the specific predominant occupational safety, health, and well-being concerns expressed by interpreters in this study, like workplace violence, loss of agency or loss of self, and organizational culture concerns, like deprioritization of self and witnessing oppression or stress from oppression. This work reiterates these minimally or previously unrecognized concerns and now, lends to their evidence base. 

Round 2

Reviewer 3 Report

Comments and Suggestions for Authors

Thanks for the changes made, although the manuscript has improved notoriously it is necessary to make small modifications

Figure 1 is super interesting, but not ve....

Title table 2, revise the format

The sections of 3.3 would be better in a table like 3.4.4, but the format should be revised, even split it... otherwise it gets longer...

I have just seen that it does not comply with an informed consent, that the authors say that there are no risks does not seem to me to be an adequate or ethical practice.

Author Response

October 18, 2024

Thank you for your willingness to evaluate another revised version of our manuscript entitled, “Occupational safety, health, and well-being concerns and solutions for management reported by sign language interpreters: A qualitative study” for publication consideration in the International Journal of Environmental Research and Public Health. Please see our responses (in red) to the additional comments detailed below.

Reviewer:

Thanks for the changes made, although the manuscript has improved notoriously it is necessary to make small modifications.

Thanks for your positive feedback about our response to reviewers’ comments in the first revision and for the opportunity to address the below small modifications.

1) Figure 1 is super interesting, but not ve....

Thanks for your feedback about how we converted the occupational safety, health, and well-being concerns of sign language interpreters from a table into a figure based on another reviewer’s comments from the first round of revisions. We are inferring from this comment that the reviewer liked the figure but that it was not very visible? Our apologies for not being sure, as this comment seems unfinished. The figure is in high resolution and the reader should be able to virtually zoom in/out for ease of viewing or the editor should be able to adjust it to a more appropriate scale, if needed.

2) Title table 2, revise the format.

We believe the reviewer would like us to make the title of Table 2 shorter, so that it is more in line with the title of Table 1. Thus, we revised the title from “Solutions for management of the occupational safety, health, and well-being concerns reported by sign language interpreters” to “Solutions for management of the concerns reported by sign language interpreters.” We hope this is more suitable.

3) The sections of 3.3 would be better in a table like 3.4.4, but the format should be revised, even split it... otherwise it gets longer...

Originally, we had both sections of 3.3 and 3.4 of our Results as tables, but another reviewer asked that we convert the table from section 3.3 into a figure, which we did (please see comment 1 above). We have placed the original table in our supplementary materials (please refer to the file entitled, “IJERPH Supplementary materials_Tables S1 and S2_Roman et al”). We parenthetically reference “Figure 1; Supplementary materials: Table S2” throughout the text of section 3.3.

4) I have just seen that it does not comply with an informed consent, that the authors say that there are no risks does not seem to me to be an adequate or ethical practice.

In the beginning of the Materials and Methods, we reported “This qualitative research study (STUDY00008720) underwent review by the University of Rochester’s Research Subjects Review Board. Based on federal and University criteria, it was deemed exempt and thus, the requirement of written informed consent from participants was waived.” Waiving written informed consent means that no signature was required, otherwise we still clearly discussed the study procedures and explanations as described in the Data collection sub-section of the Materials and Methods. At the end of the article, additional information can be found within the Institutional Review Board and Informed Consent Statements (page 21, lines 583-588). Specifically, please note that “Due to the less than minimal risk of this study, written informed consent was not required. Instead, an information sheet was reviewed with all participants in advance to beginning data collection.” The only difference between an information sheet and a consent form is that there is no signature field. We state that there was “less than minimal risk” opposed to “there are no risks,” as this comment indicates.